# Current Soil Degradation Assessment in the Thua Thien Hue Province, Vietnam, by Multi-Criteria Analysis and GIS Technology

Son Hoang Nguyen [1,2], Dan Ngoc Nguyen [1], Nhung Nguyen Thu [3], Hai Hoang Pham [3], Hang Anh Phan [4,*] and Cham Dinh Dao [3]

1    Faculty of Geography, University of Education, Hue University, 34 Le Loi Street, Hue City 530000, Thuathienhue Province, Vietnam; nhsonsp@hueuni.edu.vn (S.H.N.); nguyenngocdan@hueuni.edu.vn (D.N.N.)

2    Institue of Open Education and Information Technology, Hue University, 05 Ha Noi Street, Hue City 530000, Thuathienhue Province, Vietnam

3    Institute of Geography, Vietnam Academy of Science and Technology, 18 Hoang Quoc Viet, Cau Giay, Hanoi 100000, Vietnam; nguyenthunhung@ig.vast.vn (N.N.T.); phamhoanghai@ig.vast.vn (H.H.P.); ddcham@ig.vast.vn (C.D.D.)

4    Faculty of Geosciences, University of Science, Hue University, 77 Nguyen Hue Street, Hue City 530000, Thuathienhue Province, Vietnam

*    Correspondence: pahang@hueuni.edu.vn

**Abstract:** This article aims to provide a scientific basis for solutions to use soil cover for sustainable agricultural and rational forestry development. We used traditional methods such as survey and data collection; soil profile comparison method; vegetation indicator for soil degradation; determining the physical and chemical limiting factors of the soil; combined with the application of Geographic Information Systems (GIS) technology and the multi-criteria method (MCE) to conduct a soil degradation assessment for the Thua Thien Hue province, Vietnam. In this study, nine indicators affecting the current soil degradation were selected and then the single-dimensional maps were superimposed to form the current soil degradation map for the study area at a scale of 1/10,000. The study results show that: lightly degraded soil accounts for 48.81% of the total natural area of the study area; medium degradation accounts for 22.07%; and severe degradation accounts for 19.66%. The study results show that most of the soil in the study area is at a moderate to severe level of degradation and shows the need for synchronous implementation of reasonable solutions to prevent degradation and use soil sustainably in the Thua Thien Hue province, Vietnam.

**Keywords:** degradation; current degradation; Thua Thien Hue province; GIS; multi-criteria analysis (MCA)

## 1. Introduction

The United Nations Convention to Combat Desertification (UNCCD) recognizes that soil degradation is an environmental issue that needs attention. Severe soil degradation has threatened over 900 million people in about 100 countries, accounting for 25% of our planet's land area [1]. The worst situation is in Africa, where up to 73% of arable land has already been exhausted, and up to 66% of the continent is either desert or arid. About 800 million people living in arid regions suffer from hunger [2]. The above problems show that land degradation contribute to a 40–75% reduction in agricultural land productivity in the world.

Soil degradation is often understood as the decline in biological productivity of the soil system due to natural factors and is more severe due to human impacts [3]. Soil degradation in drylands is often referred to as desertification [4]. Soil degradation occurs due to many reasons, such as human impact and related physical and chemical environmental

factors, but mainly due to human misuse of land as an essential leading factor [5]. In addition, due to the pressure of population explosion, the urbanization process's negative impacts have caused the agricultural production area to be increasingly narrowed, and land resources have been fully exploited without good fostering. As a result, the soil will become increasingly degraded in terms of physical, chemical, and biological properties if there are no reasonable and timely interventions. In [2], a study on Soil Degradation Assessment and Livestock Perception in the Borana Area of Southern Ethiopia, soil degradation was defined as: a loss of vegetation cover leading to the soil surface being quickly exposed to the wind, increasing the process of soil erosion and leaching the organic components that bring vitality to the growth of plants [6]. In the work of [7] on the Analysis of Herbaceous Flora in Lohi Bher Wildlife Park under changing environmental pressures, it is also confirmed that soil degradation is the retreat (degradation) of the vegetation cover [7]. It includes changes in species composition, loss of biodiversity, reduced biomass production, and increased risk of soil erosion. In the work of [8], it was demonstrated that the vegetation cover in Libya had changed both qualitatively and quantitatively. This is due to many influencing factors, such as rainfall, overgrazing, improper land use management, frequent droughts, seasonal fire outbreaks, and soil erosion [8,9].

Currently, soil in the central region of Vietnam, including the Thua Thien Hue province, is facing degradation and a high risk of desertification. Soil degradation and the risk of land desertification in the Central region are mainly due to erosion and leaching, which occurs most strongly in the rainy season and is concentrated in mountainous and semi-mountainous areas [10]. The area of the Thua Thien Hue province has a high average annual temperature (24–25 °C) and a relatively large daily and annual temperature range, promoting the process of tropical weathering. Physico-chemical weathering occurs strongly in mineralization, and the decomposition rate of organic matter is high. The average annual rainfall is significant (over 2700 mm), and concentrated in the seasons; the rainy season accounts for 75% of the annual rainfall, so the process of soil erosion is widespread in steep hilly areas, as well as landslides, floods, and burials of arable land in the foothills and valleys [11]. The Thua Thien Hue province has quite a lot of poor areas, restored forests, forests without reserves, with low canopy cover (only about 30–50%), and no role in surface water regulation; only nearly 8.71% of the area has a high canopy cover of 70–80%, and 7.5% of the forest area has a canopy of 80–90%. This dramatically affects soil fertility and soil degradation [12,13]. Besides excessive exploitation and use of land for wrong purposes, excessive use of chemical fertilizers and pesticides are also the causes of soil degradation. Therefore, it is necessary to study the current soil degradation assessment for the Thua Thien Hue province, to see the extent of soil degradation and provide more scientific and practical anti-degradation measures.

## 2. Data Sources

The sources of data to develop this article include: (1) The land map of the Thua Thien Hue province, a scale of 1/100,000, provided by the Institute of Agricultural Planning and Design [13–15]. This is one of the bases for the authors to conduct field surveys to collect typical soil profiles representing ten soil groups and land use types. (2) Thematic maps, such as the map of mechanical composition and soil thickness, are extracted from the results of analysis of typical soil profiles for ten main soil groups of the study area; vegetation map (edited from 2017 forest map of the Thua Thien Hue province, Vietnam), current land use map, and bio-climatic map. (3) The physical, chemical, and biological degradation characteristics and signs detected in the field and the laboratory are considered to arrive at the current soil degradation map. (4) The documents related to the article, such as works and articles of related authors, are inherited and selected to serve the study process [11,16].

## 3. Study Methods

### 3.1. Methods of Survey and Information Collection

The survey method collects and processes data sources and documents available in the study area, including data on the current land use; thematic maps such as soil maps, vegetation maps, thick layer maps, mechanical composition maps, bioclimate maps and data sources related to the land use situation. The above documentation system has been selected, systematized, analyzed, and applied to assess the degree of land degradation in the Thua Thien Hue province and provide more scientific and practical anti-degradation measures. The specific results of normalizing input data will be analyzed in the Section 4.1.

### 3.2. Field Method to Collect Soil Profiles

Detailed survey work is carried out according to the cross-sections:

Cross section AB: from An Bang village (Vinh An commune, Phu Vang district) with coordinates $16°25'11''$ north latitude and $107°10'49''$ east longitude, along national highway 14B and provincial road 4B to the Loc Bon commune (Phu Loc), Duong Hoa (Huong Thuy), Khe Tre town, Thuong Lo (Nam Dong) to coordinates $16°11'36''$ N, $107°77'58''$ E.

Cross section CD: $16°42'2''$ N, $107°23'54''$ E along provincial road 681 through P. Hoa, Phong Thuy, Phong My, Phong Xuan communes (Phong Dien), Hong Ha, Son Thuy, Hong Thai (A Luoi) to coordinates $16°11'29''$ N, $107°11'51''$ E.

In addition, the authors conducted a survey of sub-sections: from Hue City with latitude $16°46'6''$ N, longitude $107°59'19''$ E along National Highway 49 to A Roang commune, A Luoi district with latitude $16°11'11''$ N, longitude $107°39'7''$ E (Figure 1).

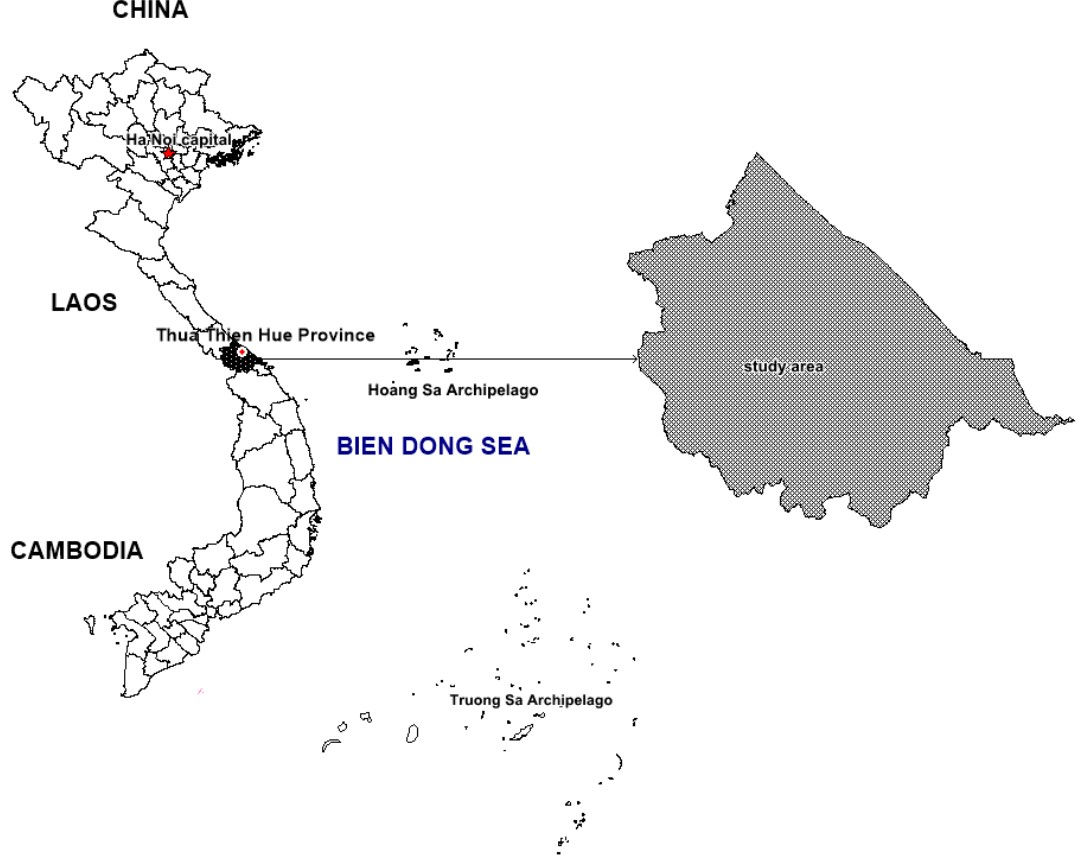

**Figure 1.** Location map of survey sites to collect profiles in the study area.

In the sections, key points are selected to study the soil profiles. There are 30 soil profiles representing ten major land groups and land use types. Detailed descriptions of the soil profiles were conducted, pictures of the profile and surrounding landscape were taken,

and soil samples were taken for analysis. Degradation forms are identified and described in the field, and degraded land units are demarcated and delineated on the base map.

### 3.3. Method of Profile Comparison

Morphological characteristics of the soil profiles reflect the degree of impact and degradation. A complete profile represents the generated soil units' level of development and maturity. The degree of soil degradation was first shown in the profile form.

Based on the soil sample, we propose the criterion of soil thickness with two main soil layers, A layer (surface layer) and B layer (accumulation layer) to serve as the evaluation criterion. In which, the layer A (surface layer) is formed by microorganisms that decompose organic substances to create porous humus, containing many nutrients; B layer (accumulative layer) is formed from dissolved and accumulated materials from the upper soil layers.

Layer A—humus accumulation layer reflects the level of humus richness of the soil, and the loss or inclusion of the A layer represents the degree of soil degradation. Some authors have used soil layers in the profile to assess the degree of erosion: slight erosion—loss of A1 layer, moderate erosion—loss of A2 layer, intense erosion—loss of B1 layer, erosion outer—exposed C layer [1,13,16].

### 3.4. Analytical Methods in the Laboratory

The collected soil samples were analyzed for chemical and physical parameters of the soil at the Institute of Geography and the Institute of Agricultural Planning and Design by the following methods: $pH_{KCl}$ analysis by pH met method; analysis $Ca^{2+}$, $Mg^{2+}$ by measuring AAS—Atomic adsorption spectroscopy; organic matter analysis by Walkey Black method; analysis of total N by Kendan method; analysis of easily digestible $K_2O$ and $P_2O_5$ by colorimetric method; analysis of total $K_2O$ and $P_2O_5$ by HF, HCl, $HClO_4$ destructive methods; the mechanical components were analyzed by the Robinson method.

### 3.5. Multi-Criteria Analysis (MCA)

This method was performed according to the following steps:

Step 1: Determine the importance of the factors. Analytical Hierarchy Process (AHP) can make decisions, arrange the order of the criteria to be considered and, thereby, make the most reasonable decision [17,18]. Based on the research experience of experts in previous studies, the factors' comparative values will be assigned according to the priority comparison scale [19] (Table 1).

**Table 1.** Scale to compare the importance of factors.

| Factor i Compared to Factor j | Quantitative Value |
| :---: | :---: |
| Equally important | 1 |
| More important | 3 |
| Much more important | 5 |
| Quite more important | 7 |
| Absolutely more important | 9 |
| Intermediate values | 2, 4, 6, 8 |

Table 1 shows that factors of equal importance have a quantifiable value of 1. More important factors have a quantifiable value of 3. The more important factors have a quantifiable value of 5. The very important factors have a quantifiable value of 7. The more extremely important factors have a quantifiable value of 9. The intermediate value factors will have a value of 2, 4, 6, and 8.

Step 2: Normalize the matrix. To normalize the importance matrix of the criteria, we divided the value of each cell in the column by the total value of that column. The weighted mean (Wi) was calculated, which was calculated by taking the sum of the weights of the factor Xi relative to Xj after normalization divided by n. To determine the reliability

of the weight (Wi), it was necessary to calculate the consistency index CR (Consistency ratio), CR < 0.1, to satisfy the consistency condition. The formula for calculating CR was as follows:

$$CR = \frac{CI}{RI} \text{ with } CI = \frac{\lambda - n}{n - 1} \tag{1}$$

$$RI = \frac{CI_1 + CI_2 + \cdots + CL_n}{n} \tag{2}$$

$$\lambda max = \frac{1}{n} * \left( \frac{\sum_{n=1}^{n} W_{1n}}{W_{11}} + \frac{\sum_{n=1}^{n} W_{2n}}{W_{12}} + \frac{\sum_{n=1}^{n} W_{nn}}{W_{11}} \right) \tag{3}$$

RI (Random index) is the random index (Table 2); λmax is the eigenvalue of the matrix.

**Table 2.** Standardized Random Index (RI) values of mean random consistency index.

| Hierarchy Matrix | 1 | 2 | 3 | 4 | 5 | 6 | 7 | 8 | 9 | 10 |
|---|---|---|---|---|---|---|---|---|---|---|
| RI | 0 | 0 | 0.58 | 0.90 | 1.12 | 1.12 | 1.32 | 1.41 | 1.45 | 1.49 |

Calculate Si value: The total S value will be calculated according to the formula:

$$S = \sum (W_i x X_i) \text{ with } i = 1 \dots n$$

Step 3: Hierarchy of total S values. Using the reclassification method in GIS and the regression algorithm in Excel to classify the total S values [20] according to each other value range depending on the research content.

### 3.6. Expert Solution

This method was implemented based on the knowledge and experience of scientists and experts related to the study fields based on direct exchange and consultation. The contents of the consultations were priority order and pairwise comparison between indicators to assess the status of land degradation. Eight experts and scientists were consulted, including 3 experts and scientists from the Institute of Geography, Vietnam Academy of Science and Technology, which had many studies on natural disasters related to the Thua Thien Hue province, 4 scientists from the Hue University and an expert from the Department of Natural Resources and Environment of the Thua Thien Hue province. The results of the consultation on "The level of agreement with the priority order and comparison of pairs of indicators to assess the current status of land degradation" indicate that the level of complete agreement and agreement accounts for 87.5% of the responses from consulted experts and scientists. The synthesis and analysis of the profound and scientific suggestions from experts had contributed to strengthening the scientific foundation and the results of assessing the current soil degradation in the Thua Thien Hue province.

### 3.7. Method of Building Soil Erosion Map

Using the universal soil loss equation

$$A = R \times K \times LS \times C \times P$$

where: A is the average amount of soil loss due to erosion (tons/ha/year); R is the index of rain erosion; L is the slope length index; S is the slope index; K is the index of soil erosion resistance; C is the vegetation cover index; P is the soil protection index of the anti-erosion measures.

### 3.8. Methods for Processing and Editing Maps (GIS)

GIS supports building and managing data and establishing thematic maps. The hierarchical analysis model will support GIS, synthesize information, and assign the most

appropriate weights to the factors [13,21]. After decentralization, weighting factors, using ArcGIS 10.3 software to overlay, combined with statistics, and integrate weights to obtain results on soil degradation (Figure 2).

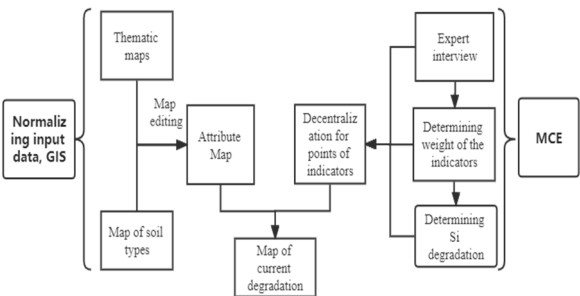

**Figure 2.** Diagram of implementation steps.

## 4. Study Results

*4.1. Normalizing Input Data*

4.1.1. Soil Data

We standardize the land map data in the same format as the built-up soil degradation types at the scale of 1/100,000 (Figure 3); the vector structured data shows the land demarcation lines with the property's information such as soil type, mechanical composition, layer thickness, and physico-chemical characteristics of the soil based on collecting typical profiles representing ten main soil groups in the study area, including:

Red and yellow soil group (F): The red and yellow soil group has the largest area, with 352,880.57 ha, accounting for 70.11% of the province's natural area, distributed in almost all districts; (2) Gray soil group (X): This group has a small area, accounting for 0.04% of the province's natural area. This type of soil is distributed only in the Phong Dien district; the soil is usually red-yellow, has mechanical components from sandy to heavy, and has good permeability and water storage; (3) Red-yellow humus in the mountains (H): accounting for 2.85% of the province's natural area. They were distributed at over 900 m in Phu Loc, Phong Dien, Nam Dong, and A Luoi districts; (4) Sandy soil group (C): Accounting for 9.48% of the province's natural area, including two types of white dune soil and sandy sea soil. This soil group is formed in coastal areas and estuaries, strongly influenced by samples and parent rock; (5) Saline soil group (M): Saline soil consists of two types: high, low, and medium (Mn, M) saline soils, accounting for only 1.58% of the province's natural area; however, saline soils are present in most of the seaside districts; (6) Acidic soil group (S): Acidic soil group accounts for 1.03% of the natural area of the province, distributed mainly in Quang Dien, Phu Vang, Phu Loc, Phong Dien, and Huong Tra districts. The morphology of the soil profile is differentiated, with the topsoil usually having a dark gray color, followed by the alum layer with yellow spots and red spots, and, in some places, tubular formations are also encountered. The mechanical components of the soil vary widely depending on the source of the accretion material; (7) Alluvial soil group (P): Alluvial soil group has an area of 37,518.67 ha, accounting for 7.45% of the province's natural area, distributed in almost all places in the Thua Thien Hue province; (8) Swamp and peat soil group (J and T): This group accounts for 0.02% of the province's natural area, distributed in the Phong Dien and Phu Loc districts. In the profile, there is no A-layer, the structure is unknown, and the whole view is firmly spread. Wetlands are often rich in organic matter, the reaction of the soil is very acidic, and the soil contains many toxic deoxidizers for plants, such as $Fe^{2+}$ and $H_2S$. Soil is poor in phosphorus and potassium, has slow organic matter decomposition, and has a light mechanical composition; (9) Valley land due to slope products (D): Accounting for 0.11% of the province's natural area, distributed in the Phong Dien district. The profile of the valley soil due to sloping products is black, dark gray, and dark brown in the surface layer, which changes suddenly to the white sand layer below. The lower layers are yellow-brown, and gravel accumulates; (10) Rocky inert

erosive soil €: Accounting for 0.99% of the province's natural area, distributed in Huong Thuy and Phu Loc districts, formed on shale rock samples and due to many different causes but mainly due to human impact, the soil becomes hardened, laterite clumps exist, and the nutrient content is low.

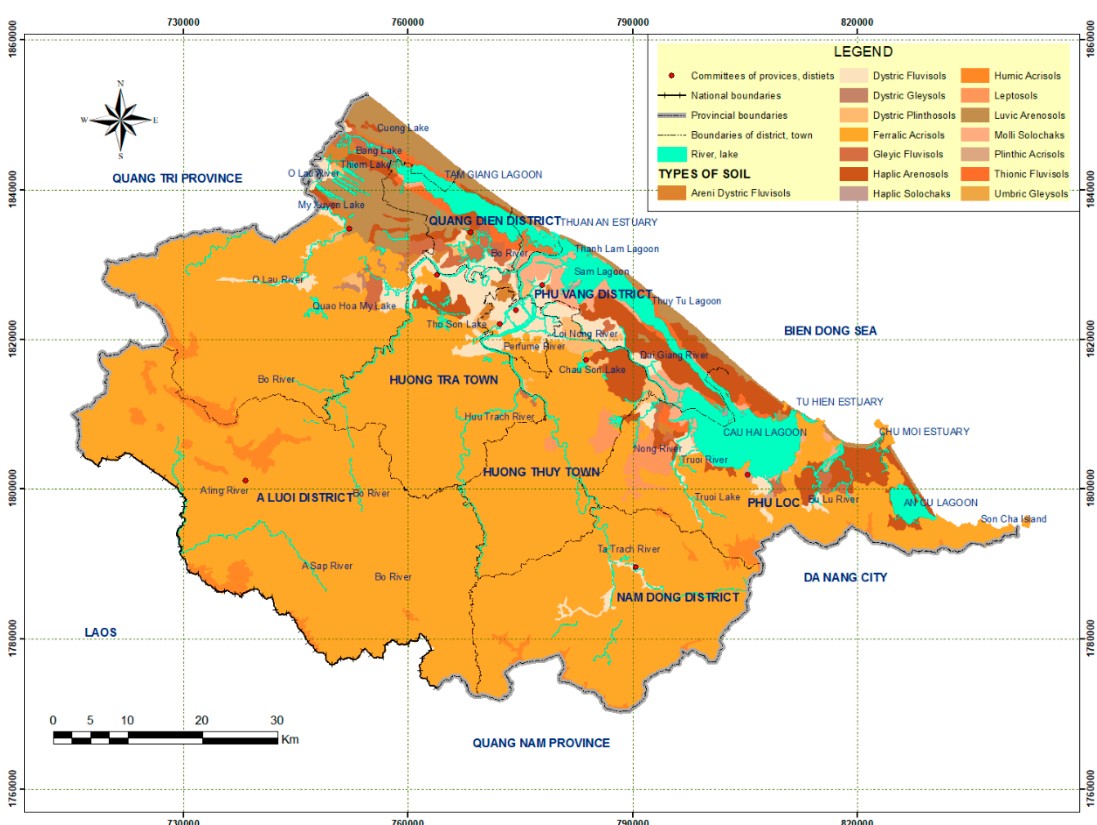

**Figure 3.** Land map of the Thua Thien Hue province, Vietnam.

### 4.1.2. Degradation Type Data

The main factors affecting the current soil degradation include chemical degradation, physical degradation, biological parameters, and current erosion. These factors interact, affecting and determining the area's current level of soil degradation.

a.    Group of indicators of chemical degradation

The analysis of the physical and chemical properties of some typical soil units of soil groups shows that most soils' humus content is inferior. The sandy soil group has the lowest humus content (0.17–0.28%); the average minimum value is only 0.05%. Soil groups are distributed in low-lying terrain, only alluvial soil covered with sea sand has average humus content (2.2%), and other soils are poor. In the red-yellow soil group, the red-yellow soil on metamorphic clay has the highest average humus content (2.86%), red-yellow soil on ancient alluvium (2.4%), and other soils from 0.84–1.73%.

In terms of acidity, a commonly used value for determining soil acidity is $pH_{KCl}$. The degenerative process increases the acidity of the face or the entire anatomy. In forest soil, $pH_{KCl} = 4.5–5.5$; when degraded, $pH_{KCl} = 3–4$. Analysis of soil samples in the study area showed that most soils have $pH_{KCl} < 5$; only grey alluvial soil, alluvial soil in streams, alluvial soil covered with sea sand, and red and yellow soil on clay rock had a $pH_{KCl} < 4.5$.

In terms of total phosphorus content ($P_2O_5$), all soils in the study area have average $P_2O_5$ content ranging from very poor to poor, from 0.009 to 0.083%, the highest is grey alluvial soil and red-yellow humus soil on metamorphic rocks, the lowest type of gray soil on acid igneous rocks. Sea sand soil has the lowest $P_2O_5$ minimum (0.002%), and red-yellow soil on acidic magma has the highest maximum $P_2O_5$, which is rich in phosphorus (0.196%)

in the alluvial soil group, from 0.013 to 0.083%. The red and yellow soil groups ranged from 0.017% to 0.075%. The groups of saline soils, alkaline soils, valley soils due to sloping products, and inert erosive soils all have $P_2O_5$ from very poor to poor (0.023–0.09%).

The easily digestible phosphorus content of soils in the study area is inferior (0.78–7.33 mg/100 g of soil). Alluvial soil covered with sea sand has the highest average phosphorus content (7.17 mg/100 g of soil, and the maximum mean value is 13.3 mg/100 g). The red and yellow soil groups, all of which have the above values, are inferior.

The cation exchange capacity of the soil (CEC)

The change of CEC and the composition of cations of each soil varies significantly from 1 to 100 ldl/100 g of soil. The CEC value < 1 ldl/100 g of soil indicates minimal adsorption capacity of the soil. Most soils have medium CEC, incredibly steep valley soils have high CEC (28.28 ldl/100 g), and the lowest is alluvial soil (4.18 ldl/100 g).

b.  Group of indicators of physical degradation

Criterion of soil layer thickness (Figure 4).

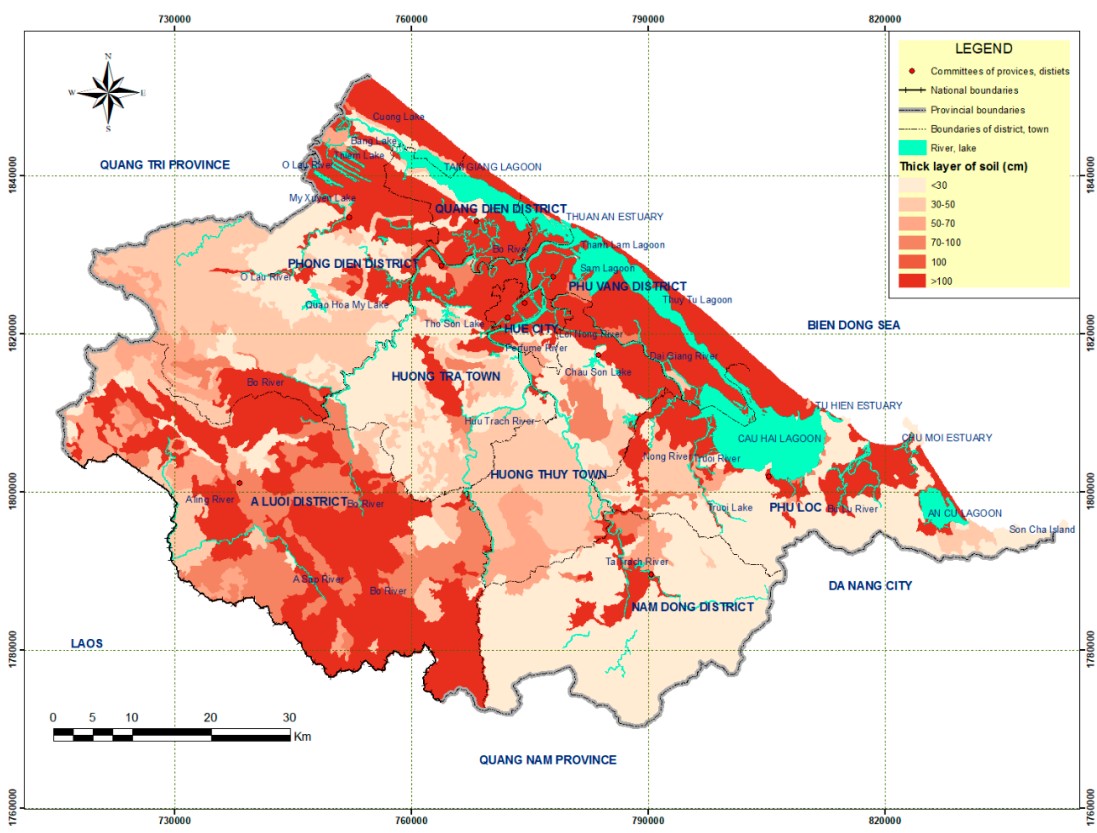

**Figure 4.** Soil thickness map of the Thua Thien Hue province, Vietnam.

The results of analysis of physical and chemical properties of some typical soil units of soil groups show that soil with a thickness of less than 50 cm has the largest area, accounting for 47.21% of the total natural area, generally in types of soil such as red, yellow soil on acid magma, and light-yellow soil on sandy rock. Soil with a thickness of over 100 cm accounts for 32.41% of the total natural area, found in sandy soils, alluvial soils, saline soils, red soils, yellow on metamorphic clay, and red-yellow humus. Soil with medium thickness at least accounts for 14.02% of the total natural area in almost all soil types in the province.

Indicators of mechanical components of soil (Figure 5):

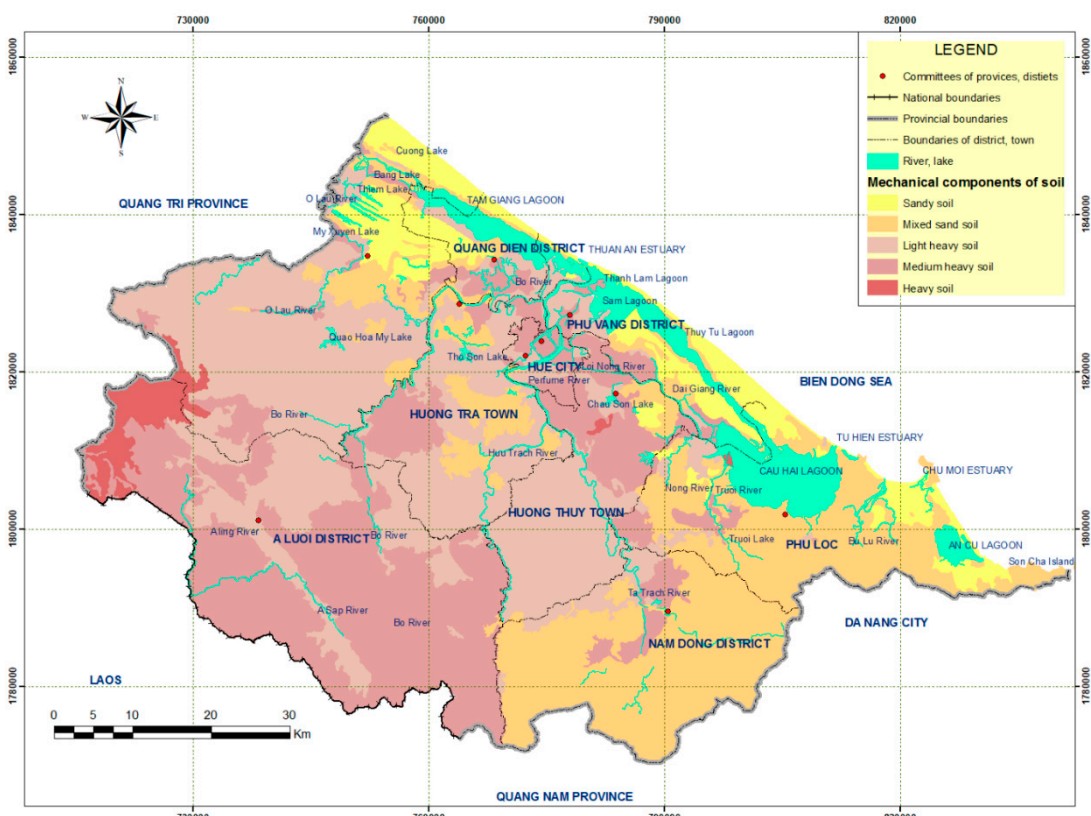

**Figure 5.** Map of mechanical components of soil in the Thua Thien Hue province, Vietnam.

The results of mechanical components analysis of some soil types in the study area show that there are differences between soil types.

The group of sandy soil, and gray soil on acid magmatic rock, has the lightest mechanical components, and sand grain grade predominates with 80–97%. Therefore, the ability to hold the soil's water, nutrients, and buffering properties is inferior. The group of alluvial soils with mechanical components fluctuates from sandy to clay-heavy soil, in which alluvial soil has a red-yellow patchy layer with demonic composition predominating with 44.77%. The humus group has mechanical components from sandy to clay. The red-yellow soil group has mechanical components that vary from sand mixed with clay-heavy soil.

Drought indicator (Figure 6)

The drought indicator is calculated from the ratio of precipitation to potential evaporation that FAO has proposed to assess the risk of desertification. This ratio is equal to 1 at the onset of drought, and from 0.05 to 0.65 is the appearance of a desert. For the study area, to assess potential soil degradation, the drought index is based on the calculation of the length of the dry season (months with rainfall < 100 mm) and the number of dry months (months with rainfall < 25 mm) selected and graded.

c.   Signs of vegetation indicate soil degradation (Figure 7)

In addition to helping to identify degraded soil, vegetation also reduces the risk of soil degradation [20]. There is a very close relationship between vegetation cover and soil degradation levels [21]: the soil under the evergreen forest is very stable and has high coverage, so the soil is less eroded and washed away. In the transition zone from the sea to the plain, the forest has contributed to the fixation of sand strips and dunes, forming a natural corridor to break waves and sand, and improving the soil cover in the direction towards the sea, limiting coastal erosion effectively (Table 3).

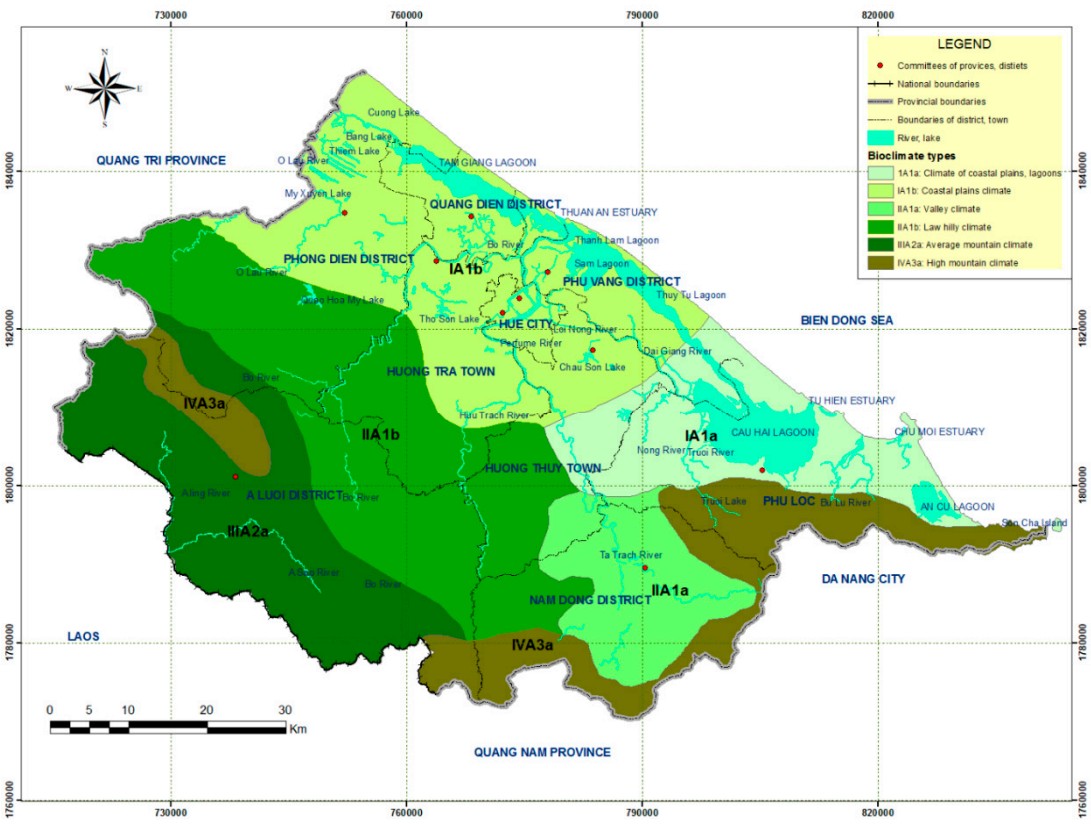

**Figure 6.** Bioclimatic map of the Thua Thien Hue province, Vietnam.

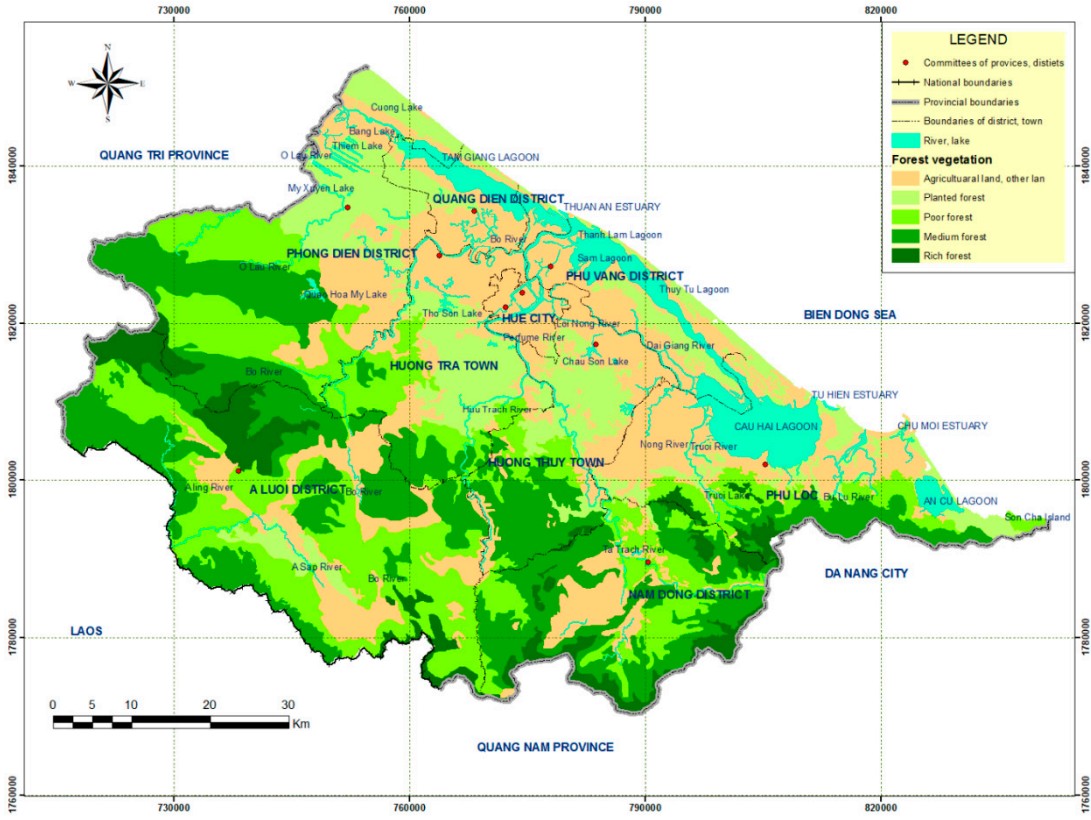

**Figure 7.** Forest vegetation map of the Thua Thien Hue province, Vietnam.

**Table 3.** Degradation type hierarchy in attribute data.

| | Hierarchy | Characteristics | Symbol | Level of Degradation |
|---|---|---|---|---|
| Erosion (ton/ha/year) | Weak erosion | <100 | $XM_1$ | Poor |
| | Medium erosion | 100–500 | $XM_2$ | Medium |
| | Strong erosion | >500 | $XM_3$ | Strong |
| | | Number of dry months | | |
| Drought (month) | Mild drought | <2 | $KH_1$ | Poor |
| | Average drought | $\geq$2–4 | $KH_2$ | Medium |
| | Severe drought | >4 | $KH_3$ | Strong |
| Soil thickness (cm) | Thick soil layer | >100 | $TD_1$ | Poor |
| | Medium soil layer | 50–100 | $TD_2$ | Medium |
| | The thin layer of soil | <50 | $TD_3$ | Strong |
| Mechanical components | Heavy soil | Heavy soil mixed with clay and Limon, heavy soil mixed with clay, and clay mixed with sand | $CG_1$ | Poor |
| | Medium-heavy soil | Heavy soil mixed with lemon, Limon, heavy soil mixed with clay, and mixed with sand | $CG_2$ | Medium |
| | Light heavy soil | Sand, sand mixed with heavy soil, heavy soil mixed with sand, heavy soil | $CG_3$ | Strong |
| Vegetation | Thick vegetation | Natural forests, planted forests (with reserves) and protection forests; Land for planting perennial industrial crops and fruit trees; Two-crop rice fields, specialized land. | $TV_1$ | Poor |
| | Medium vegetation | Planted forests that have not yet closed their canopy, bamboo forests, and poor forests; Annual cropland; Land of shrubs and trees scattered; Rice field one crop. | $TV_2$ | Medium |
| | Poor vegetation | Natural grassland, vegetation on eroded soil, inert gravel; Temporary field land; Hilly land and flat fallow land, specialized land, etc. | $TV_3$ | Strong |
| Acidity | Poor acidity | $pH_{KCl} < 6$ | $DC_1$ | Poor |
| | Medium acidity | $pH_{KCl}$ 5–6 | $DC_2$ | Medium |
| | Strong acidity | $pH_{KCl} < 5$ | $DC_3$ | Strong |
| Humus content (%) | Rich humus | >3 | $M_1$ | Poor |
| | Medium humus | 1–3 | $M_2$ | Medium |
| | Poor humus | <1 | $M_3$ | Strong |
| Total phosphorus content (%) | Rich | >0.15 | $LTS_1$ | Poor |
| | Medium | 0.10–0.15 | $LTS_2$ | Medium |
| | Poor | <0.1 | $LTS_3$ | Strong |
| Total phosphorus content (%) | Rich | >15 | $LDT_1$ | Poor |
| | Medium | 10–15 | $LDT_2$ | Medium |
| | Poor | <10 | $LDT_3$ | Strong |

The drought index is based on the calculation of the length of the dry season (months with rainfall < 100 mm) and the number of arid months (months with rainfall < 25 mm) [22]. As for the erosion index criteria: The erosion map is constructed from the multiplication results of the R, K, LS, and C component maps [23,24]. To make the results more readable, erosion levels are categorized into several grades. Considering the characteristics of the research area, we chose to classify erosion levels into three grades (Table 3) [25,26]. The determination of limiting factors regarding the physical and chemical properties of the soil is based on the Vietnamese Standards of 2008 [27].

d. Current Erosion Indicator

The current erosion rate indicator most clearly reflects the current level of degradation for tropical regions such as the Thua Thien Hue province, Vietnam. The soil erosion and leaching processes are the leading causes of soil degradation, especially in areas with sloping terrain and seasonally concentrated rainfall. Therefore, the current erosion rate indicator is essential for evaluating soil degradation. According to multi-year monitoring data in the Thua Thien Hue province, on arable land with a slope of 50–80, with an annual rainfall of about 2700 mm, the amount of soil washed away on upland fields is up to 95.1 tons/ha/year. A summary of many erosion monitoring points on different slopes and land use types showed that the average annual amount of nutrients per hectare of productive land was huge: about 171 kg N; 19 kg $P_2O_5$; 337.5 kg $K_2O$; 1125 kg of organic matter [28].

The current level of soil degradation is assessed according to three levels: mild, moderate, and severe. This classification is consistent with the FAO soil degradation assessment guidelines. The criteria are selected and classified based on the preconditions leading to the current soil degradation processes in the study area (Table 3) [1,10,17].

4.1.3. Determination of the Weights and Si Degradation Values for the Indicators

The determination of the weights of the indicators by the hierarchical method is carried out in the following steps: determining the evaluation criteria, consulting experts, setting up a symmetric matrix of the evaluation results of the pairs of evaluation criteria for soil degradation in the study area, make a separate vector matrix and calculate the weighted scores of the evaluation criteria.

Determining the Xi point in the study area is based on the principle that the total Xi point of the same indicator must be 100% (so that the total Si value of an indicator is equal to the weight of that indicator) and determined according to the order to gradually increase in importance. The results show that the 10%, 20%, 30%, and 40% sets of values are suitable for each criterion and show the difference in the Si value in the following steps. The Si degradation value is calculated using Si = Wi × Xi (Table 4).

**Table 4.** System of indicators and weights for assessment of current degradation.

| The Goal of Layer A | Criteria of Layer B | | Indicator of Layer C | | | | |
|---|---|---|---|---|---|---|---|
| | Factor | Tier-1 Weight | Indicator Layer | Current Status Value | Tier-2 Weight | Rate (%) | Si |
| Current soil degradation in the Thua Thien Hue province, Vietnam | Chemical degradation | 0.06 | Soil acidity | <4–>7 | 0.15 | 10<br>20<br>30<br>40 | 0.015<br>0.030<br>0.045<br>0.60 |
| | | | Humus content | <1–>4% | 0.09 | 10<br>20<br>30<br>40 | 0.009<br>0.018<br>0.027<br>0.36 |
| | | | Total phosphorus content | <0.1–>0.15% | 0.02 | 10<br>20<br>30<br>40 | 0.002<br>0.004<br>0.006<br>0.008 |
| | | | Easy-digesting Phosphorus | <10–>15% | 0.04 | 10<br>20<br>30<br>40 | 0.004<br>0.008<br>0.012<br>0.016 |
| | | | Ion exchange capacity in the soil | <10–>40 meq/100 g soil | 0.02 | 10<br>20<br>30<br>40 | 0.002<br>0.004<br>0.006<br>0.008 |

**Table 4.** *Cont.*

| The Goal of Layer A | Criteria of Layer B | | Indicator of Layer C | | | | |
| --- | --- | --- | --- | --- | --- | --- | --- |
| | Factor | Tier-1 Weight | Indicator Layer | Current Status Value | Tier-2 Weight | Rate (%) | Si |
| Current soil degradation in the Thua Thien Hue province, Vietnam | Physical degradation | 0.10 | The thick layer of soil | 0–>150 cm | 0.06 | 10<br>20<br>30<br>40 | 0.006<br>0.012<br>0.018<br>0.24 |
| | | | Mechanical components | Particle level <0.02–2 | 0.04 | 10<br>20<br>30<br>40 | 0.004<br>0.008<br>0.012<br>0.016 |
| | | | Drought | <2–>4 months | 0.02 | 10<br>20<br>30<br>40 | 0.002<br>0.004<br>0.006<br>0.008 |
| | Plant | 0.61 | Plants | Plant | 0.36 | 10<br>20<br>30<br>40 | 0.036<br>0.072<br>0.108<br>0.144 |
| | Current erosion | 0.23 | Current Erosion | 0–>50 tons/ha | 0.18 | 10<br>20<br>30<br>40 | 0.018<br>0.036<br>0.054<br>0.072 |

### 4.2. Current Degradation Synthesis Results

The assessment results show that the current soil degradation in the Thua Thien Hue province, Vietnam has three levels: mild, medium, and severe. Each level has different manifestations in appearance regions and dominant degenerative processes (Table 5, Figures 8 and 9).

**Figure 8.** Current degradation map in the study area.

**Table 5.** Existing scale and extent of degradation in the study area.

| Level of Degradation | No Soil Unit (Plot) | Scale (ha) | Ratio (%) | Symbol |
|---|---|---|---|---|
| Poor degradation | 462 | 245,724.72 | 48.81% | HT1 |
| Medium degradation | 521 | 111,146.09 | 22.07% | HT2 |
| Severe degradation | 332 | 98,987.36 | 19.66% | HT3 |

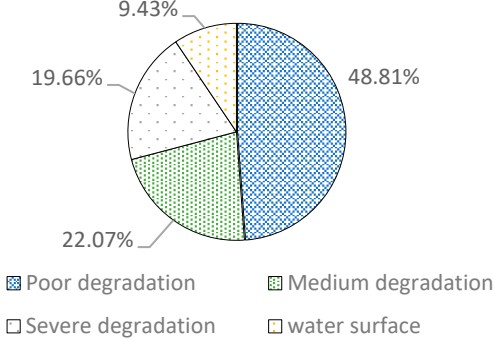

**Figure 9.** The current rate of degradation in the study area.

Mild or non-degenerative (HT1): accounting for 48.81% of the natural area of the study area. Most of this area is under natural forest cover and has been positively impacted by humans, so most of this area has not shown any signs of degradation, mainly distributed in A Luoi, Nam Dong, and areas surrounding the Bo river and Huong river. The profile is characterized by the O and A layers of humus; the soil layer is over 150 cm thick. The mechanical components are mainly medium to heavy flesh. The soil is moist, porous, and has many roots and animal burrows.

Chemical properties of HT1: medium to rich humus content and medium to rich nutrient content. $Mg^{2+}$, $Ca^{2+}$, $K^+$, and $Na^+$ exchange cations are poorly moderate. Nutrient elements N, P, and K show richness.

Moderate degradation (HT2): accounting for 22.07% of the province's natural land area, distributed mainly in the districts of Phong Dien, A Luoi, Huong Tra, Nam Dong, and Phu Loc, with the current status of land use as an area for annual crops, scattered shrubs, and trees, mixed gardens. These soils show signs of average decline in fertility and signs of physical, chemical, and biological decline compared to the soil from which they arise. The soil profile HT2 usually has a thin to medium humus layer and the structure is no longer in a primeval state. The surface layer structure is broken and discolored. In the profile, small stone fragments and a few gravel stones often appear. The soil thickness is usually from 50 to 100 cm. The mechanical components of HT2 are usually lighter than HT1. The clay grain-grade composition of the surface layer usually decreases significantly with the deep layer. The structure is broken in the surface layer with the content of fine particles and water solubility increased.

Severe degradation (HT3): severe degradation accounts for 19.66% of the province's natural land area, distributed mainly in dunes, sandy grasslands, and on inert erosive and rocky soil in the coastal districts of Phong Dien, Quang Dien, and Phu Loc. A large part of this area appears under the current land use status as shifting cultivation, unused land, scattered trees, shrubs, and newly planted forests in Nam Dong, A Luoi, and Huong Thuy districts. The soil profile shows a poor soil layer and destroyed soil structure. The soil's mechanical components increase the source of raw materials or have a physical barrier. The humus layer is almost absent, and the proportion of slags and gravels increases. The nutrient base of HT3 soils is usually at a poor limit for crops. However, in some places, the humus content of the top layer is still better (>3%), but the humus reserve is minimal. In the soil profile, gravel appears in the shallow layer, from the surface to a depth of about 30 cm. Many dark-brown clumps appear with a diameter of 0.5–3.0 mm. At the soil layer from 30–45 cm, there are many small- to medium-sized gravels with many black

and tough clumps. Severely degraded soil units undergo strong chemical and physical degradation processes.

Thus, in the study area, degraded soil is currently at a high level of medium and heavy. These two levels account for over 40% of the total natural area. The current degradation levels are distributed in the districts as shown in Table 6 below:

**Table 6.** The current scale of soil degradation by districts in the study area.

| Administrative Unit | Level of Degradation | | | River. Pool (ha) | Total Area (ha) |
|---|---|---|---|---|---|
| | TN1 (ha) | TN2 (ha) | TN3 (ha) | | |
| Phong Dien | 45,370.81 | 29,092.02 | 20,258.99 | 7702.30 | 101,274.1 |
| Phu Loc | 25,210.47 | 12,360.96 | 20,217.06 | 12,998.96 | 69,687.42 |
| Nam Dong | 39,246.35 | 16,274.24 | 7470.34 | 192 | 63,224.93 |
| A Luoi | 78,956.93 | 22,574.88 | 20,258.99 | 1554.76 | 123,295.6 |
| Huong Tra | 17,156.13 | 17,908.84 | 10,499.94 | 6327.67 | 51,569.58 |
| Huong Thuy | 27,080.22 | 5312.59 | 8364.72 | 5130.22 | 44,697.75 |
| Phu Vang | 8403.97 | 2274.15 | 9185.83 | 10,190.38 | 28,845.33 |
| Quang Dien | 3085.16 | 4731.09 | 4212.14 | 4493.11 | 14,831.5 |
| Hue City | 1214.65 | 617.33 | 4439.96 | 980.83 | 6252.16 |
| Total | 245,724.7 | 111,146.1 | 98,987.36 | 47,820.23 | 503,678.4 |
| Ratio (%) | 48.81 | 22.07 | 19.66 | 9.43 | 100 |

## 5. Discussion

The use of traditional methods such as survey collection and data analysis, and modern methods such as GIS and multi-criteria analysis to determine the scale, distribution, and level of soil degradation is considered to be the first step to providing solutions to prevent and overcome soil degradation for the Thua Thien Hue province, Vietnam. Applying the GLASOD model (Global Assessment of Soil Degradation) allows for adjustment of the indicators of soil degradation in the study area, such as degradation patterns, degradation levels, and causal factors, demonstrating the model's flexibility.

The study results shown on the map have confirmed that more than 40% of the land area in the Thua Thien Hue province is in moderate and severe degradation. The current factors causing soil degradation in the study area have been identified, in which humanity is the main factor. This is the result of unreasonable land use. The primary forms of soil degradation are caused by humans, such as soil being washed away and degraded by water. This results from intense and unreasonable land exploitation for a long time, incredibly inappropriate use and protection of forests due to low cover, and heavy seasonal rains, such that erosion is taking place firmly on fertile, sandstone, acid magma, shale, and inert erosive soils. In addition, soil pollution is also a significant cause of soil degradation. It is a consequence of using chemical fertilizers and pesticides, industrial, handicraft, aquaculture waste, and chemicals used in war and civil activities.

This assessment highlights the need for rapid action to minimize soil degradation. Many issues need special attention to prevent and limit soil degradation in the Thua Thien Hue province. In the immediate future, the following solutions should be prioritized:

### 5.1. Non-Structural Solutions

5.1.1. Solutions on Management Policies, Laws, and Education Propaganda: Raising Awareness of All Levels and Communities in the Study Area

It is necessary to develop and issue appropriate policies on land allocation and forest allocation, regulations on management, and the use of different types of land. It is also necessary to include sustainable land use models into the pilot to increase the effectiveness of education and propaganda. Sustainable land use models have been successfully applied by Dao, D.C., Nguyen, H.S., Nguyen, T. Q., and Phan, A. H in the riverside coastal areas of Quang Binh province, Vietnam. In particular, it is necessary to open short- or medium-term training courses depending on the scale from the province to the village according to the

maxim of holding hands, updating information, and actively training local officials with priority policies of the province or through programs and projects.

### 5.1.2. Solutions on Planning and Restructuring of Crops and Livestock That Are Suitable from an Economic-Ecological Point of View and Sustainable Development of Land Resources

Mountainous areas: Implementing sedentary cultivation, habitation, protection, and forest development is required to ensure food security, avoid and restrict soil degradation, etc. In high mountains and hilltops, it is necessary to protect watershed forests, bare green hills, and bare hills to keep and regulate water sources, and limit soil erosion and wash away. Places with steep slopes and thin soil layers need to be planted in tapes, on top of which shrubs should be planted. In the next belt, the pine, Manglietia insignis, Acacia mangium Willd, Acacia auriculiformis A. Cunn. Ex Benth, etc., should be planted. The plants, such as fruit trees and short-term industrial plants, should be planted at the bottom. Moreover, it is necessary to actively plant more new forests with trees such as acacia, casuarina, and pine, with the maxim that forest land must have an owner.

Hilly areas: It is necessary to develop the hill garden economy on a reasonable scale. Currently, banana, cinnamon, pepper, orange, and tangerine trees are suitable for the ecological conditions of the region and bring high economic efficiency. Proposed agro-forestry models include Forest—Garden; Forest—Garden—Barn. In addition, it is necessary to build sustainable livelihood models based on traditional products associated with the community to help ethnic minorities in the highlands alleviate poverty and stabilize their lives: typically, the organic agriculture and animal husbandry model, the economic model of the garden, growing Ra Du rice on sloping and low land, etc. In Libya, Al-Bukhari, A., Hallett, S., and Brewer, T showed that cattle ranching has a great influence on land degradation. At the same time, the authors have also successfully evaluated and applied the organic livestock model to monitor the degradation of grass land.

The delta area: Mainly alluvial and sandy soil suitable for growing food crops (rice, maize, soybean) and food crops. Promoting intensive farming and applying technological advances in seeds, fertilizers, pest control, etc., is necessary. To form high-quality rice production areas such as the coastal Phu Loc district or develop cassava in the Phong Dien sand area to supply raw materials for processing plants.

### *5.2. Structural Solutions*

### 5.2.1. Structural Solutions for Areas Prone to Landslides

It is essential to build a new embankment and plant trees to green the area above the embankment where traffic routes and canal banks are often eroded. The area adjacent to the embankment can plant Vertiver grass because this grass variety grows fast, has good drought tolerance, requires little nutrition, and can fix the soil to limit landslides. In addition, it is also necessary to invest in the construction of embankment systems to prevent landslides in riverbanks and coastal areas.

### 5.2.2. Solutions for Irrigation Works

Solutions include designing and building more irrigation works for areas without active or lack of active water sources. We are upgrading existing lakes and dams, concreting the infield canal system.

### 5.2.3. Solutions for Combined Agro-Forestry Works

For this solution, it is possible to refer to the agro-forestry economic model (eco-village) on the deserted hills in Gio An commune, Gio Linh district (Quang Tri) by the provincial Department of Science, Technology, and Environment in collaboration with the Institute of Ecology and Biological Resources (1997–2000) to apply to hilly areas.

5.2.4. Solutions for Forestry Works

This solution includes planting watersheds, special-use forests, and greening bare land and hills. Applying farming methods on sloping land: terraced fields and drainage ditches to limit erosion and washout, planting trees to create ice to keep the soil, and digging ponds to provide irrigation water. Protective forests against flying sand and flowing sand protect the inland areas and keep fresh water sources for the production and daily life of people in the area.

*5.3. Solution in Conformity with the Degradation Levels of Land Units*

5.3.1. Mild or Non-Degenerative Unit (HT1)

The leading solution for lightly degraded land units is to conserve, rationally exploit land resources, invest in intensive farming, and regularly assess soil fertility to prevent degradation. Constructing new, improving, and repairing damaged irrigation works is essential to develop small- and medium-sized irrigation systems. Intercropping techniques should be used, and short-day plants should be rotated with long-day plants. Ethnic minorities in the highlands can be helped to end poverty and stabilize their lives by developing sustainable livelihood models based on traditional goods and linked to the community.

5.3.2. Light to Moderate Soil Degradation Unit (HT2)

Due to the degree of soil degradation from mild to moderate, the solutions are somewhat more favorable than the heavily degraded unit (HT3). For these lands, it is necessary to continue improving the soil, prioritizing testing, and planting trees of high economic value to ecological conditions. Also, protecting the environment, preventing erosion, and washing away the soil are necessary. Afforestation still needs to be maintained; moreover, it is necessary to increase investment in industrial plants of high economic value, such as coffee, pepper, and fruit trees. Afforestation can be combined with conservation and tourism development activities (natural reserve of Bach Ma National Park, Lang Co Bay, Canh Duong, and Thuan An beaches).

5.3.3. Serious Soil Degradation Unit (HT3)

The degraded nature of HT3 is severe because the soil is currently degrading at a very high level. At the same time, there are no timely measures to overcome it. The potential risk from the potential for degradation is still present and can have a synergistic effect with existing degradation to create soil degradation that is as dangerous as other natural hazards. Therefore, the first thing to do is improve the soil and start planting. Severely degraded land in mountainous areas should be prioritized for investment in reforestation, zoning, and protection of existing forest areas, promoting land and forest allocation to households and forestry enterprises. Reforestation should also be considered in hilly and coastal sandy areas, but structural transformation can be studied. Using lands such as replicating ecological and economic models on sandy and hilly areas has brought economic benefits to people, promoting aquaculture, investing in tourism development, etc.

**6. Conclusions**

We utilized cutting-edge methods such as GIS technology, surveying, data collecting, comparison of land profiles, a vegetation indicator for soil degradation, restrictions on soil chemistry, and physics, along with surveying (determined by the VN-2008 Standard). Based on domestic and foreign studies, the authors proposed to select nine indicators affecting the current soil degradation in the study area, then proceed to superimpose single-dimensional maps to establish the current soil degradation map for the study area at a scale of 1/100,000. The results have determined the current level of soil degradation in the study area as follows: Mild degradation prevails in 48.81% of the natural area of the study area. Most of this area is under natural forest cover and positively impacts humans; The average degradation accounts for 22.07%, mainly distributed in Phong Dien, A Luoi, Huong Tra, Nam Dong, and Phu Loc districts. Severe degradation accounts for 19.66%,

mainly distributed in the coastal districts of Phong Dien and Quang Dien and mountainous districts such as Nam Dong, A Luoi, and Phu Loc. The study results are the basis for further studies on the synthesis of soil degradation in the study area and other areas. It is also the basis for proposing solutions to prevent and overcome the consequences of soil degradation. The structural and non-structural solutions and specific solutions for the degradation levels in the study area have been proposed based on the area's land-use planning, geographical and soil views, views on the regions of land generation and degradation, the views on the types of soil degradation and the views on the degrees of soil degradation, and the results of the integrated soil degradation assessment.

In the future, the issue of land degradation should be expanded to other provinces/ cities, or further studied in districts/communes of the Thua Thien Hue province; to assess the status and degradation of land, we propose measures to improve and use land reasonably, serving the effective development of economic programs such as the National Program, one commune—one product (OCOP).

**Author Contributions:** Conceptualization, S.H.N. and D.N.N.; methodology, S.H.N., C.D.D. and H.A.P.; software, C.D.D.; validation, S.H.N. and N.N.T.; formal analysis, C.D.D.; investigation, C.D.D. and H.A.P.; resources, S.H.N. and H.H.P.; data curation, H.A.P.; writing—original draft preparation, C.D.D. and H.H.P.; writing—review and editing, S.H.N.; visualization, H.A.P.; supervision, S.H.N.; project administration, S.H.N.; funding acquisition, S.H.N. All authors have read and agreed to the published version of the manuscript.

**Funding:** This work was funded by Hue University under the Core Research Program, Grant No. NCM.DHH.2023.03 and Ministry of Education and Training under project code B2023-DHH-28.

**Institutional Review Board Statement:** Not applicable.

**Informed Consent Statement:** Not applicable.

**Data Availability Statement:** Not applicable.

**Acknowledgments:** The authors are grateful to Hue University under the Core Research Program, Grant No. NCM.DHH.2023.03 and Ministry of Education and Training for funding. The authors also acknowledge the assistance of Institute of Geography's staff during the fieldwork. The Institute of Open Education and Information Technology is greatly appreciated for providing facilities for this study.

**Conflicts of Interest:** The authors declare no conflict of interest.

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
