# Peer review of "Current Soil Degradation Assessment in the Thua Thien Hue Province, Vietnam, by Multi-Criteria Analysis and GIS Technology"

_sustainability, doi:10.3390/su151914276_

Round 1

Reviewer 1 Report

The manuscript aims to map the soil degradation for Thua Thien, Hue, Vietnam using Analytical Hierarchal Process within geospatial environment. The paper seems interesting and in the aims of the Journal. However, following are a few comments for further improvement.

1.      The writing needs improvement. Also, having a native English speaker to go over the manuscript should be warranted. For instance, unnecessary repetition of the term ‘soil degradation’ from lines 42-47 may be omitted/revised for clarity and logical flow of the sense of sentence.

2.      The introduction and other sections should be re-reviewed to avoid typing errors, for instance, refer to line 38. Also, full form of the abbreviation should be written well before the short-termed abbreviations. Like the term GIS should be clearly elaborated and abbreviated scientifically (See line 19).

3.      Introduction section should be improved by the specifically including what has been studied or known over this area and what has not been studied and how the current work contributes.

4.      Texts placed inside the Table 4 is not clear and should be thoroughly adjusted/ formatted for readability and comprehension of the readers.

5.      Spatial resolution, temporal resolution and other important characteristics of the dataset should be tabulated along with their respective source.

6.      Authors are recommended to adjust the flow diagram / flow chart for better understanding among readers. Please see the left edge of Fig. 3, which is incomplete and is being cut short.

7.      Missing line numbers in section 4, and section 4.1 (after line 194) should be updated for proper review.

8.      Resolution and quality of all the images should be improved.

9.      Discuss while analyzing the variables (input datasets) in AHP, authors performed multicollinearity check? If yes, please add relevant results in manuscript.

10.  Legends and other map elements in most of the maps, particularly in Fig. 4, 5, 6, 7, and 9 are not clear.

11.  Fig. 7 is captioned for two different maps.

12.  Please add up-to date and relevant references to support the explanation/details provided for study area in section 3.

13.  Avoid references older than 5 years (>2018)

14.  Based on your findings/conclusions, suggest a few areas for future research.

15.  Resolution and quality of all the images should be improved.

16.  Scale for all the maps covering the same extent of area should be same. Thereby, in case the spatial extent of the two boundaries is same, then same scale should be used for the two maps.

 The writing needs improvement. Also, having a native English speaker to go over the manuscript should be warranted.

Reviewer 2 Report

This study focused on the soil degradation assessment using crateria analysis and GIS method, quite novel. The paper was well organaized with convincing results. However, major revisions are still necessary before publication. My specific comments are listed as follows:

It is recommended to use the general past tense about your measurements and methology. 

Line 23, Lightly should be lightly, Line 24 Severe should be severe.

Revise the citation and reference formats to cover the requirement of this Journal.

Previous studies and corresponding comparision about the determination of soil degradation were lacked in the introduction part.

rewrite the sentence from Line 39-41.

Line 116, latitude xxx N, longitude xxx E

Figure 1 was not cited in the text. 

The distributions of soil samples should be introduced in the study area of figure 1. A figure about the field condistion of the soil samples is also recommended.

Lack of necessary legends in fiugre 1, such as the north arrow and scale bar.

Table 1 is very confusing, redraw the table, are the descriptions from Line 158 to 163 included in Table1?

I suggest a new table to offer corresponded equations 1-3, besides, incomplete dependent variable is fould in formula 3.

Define RI in Table 2.

Incomplete Figure 3

Revise the text format on the left side of Figures 3 and 4. The same problem with other figures.

Too many paragraphs in section 4.1.1

Unify the font and format of titles of all sections and subsections.

Further research of this study should be mentioned in discussion part. In addition, the improvement advise should also be mentioned.

Rewrite the conclusion section, too many descriptions are found. emerge the paragraphs in conclusion into one paragraph. 

Round 2

Reviewer 2 Report

The study has carefully revised, most of my comments have been addressed. 
